# Highly Dense FBG Temperature Sensor Assisted with Deep Learning Algorithms

**DOI:** 10.3390/s21186188

**Published:** 2021-09-15

**Authors:** Alexey Kokhanovskiy, Nikita Shabalov, Alexandr Dostovalov, Alexey Wolf

**Affiliations:** 1Novosibirsk State University, 1 Pirogova Street, Novosibirsk 630090, Russia; gefest776@gmail.com (N.S.); dostovalov@iae.nsk.su (A.D.); alexey.a.wolf@gmail.com (A.W.); 2Institute of Automation and Electrometry of the SB RAS, 1 Academic Koptyug Avenue, Novosibirsk 630090, Russia

**Keywords:** fiber bragg grating, optical fiber sensor, distributed temperature sensor, deep learning algorithms, fully connected neural network, convolutional neural network

## Abstract

In this paper, we demonstrate the application of deep neural networks (DNNs) for processing the reflectance spectrum from a fiberoptic temperature sensor composed of densely inscribed fiber bragg gratings (FBG). Such sensors are commonly avoided in practice since close arrangement of short FBGs results in distortion of the spectrum caused by mutual interference between gratings. In our work the temperature sensor contained 50 FBGs with the length of 0.95 mm, edge-to-edge distance of 0.05 mm and arranged in the 1500–1600 nm spectral range. Instead of solving the direct peak detection problem for distorted signal, we applied DNNs to predict temperature distribution from entire reflectance spectrum registered by the sensor. We propose an experimental calibration setup where the dense FBG sensor is located close to an array of sparse FBG sensors. The goal of DNNs is to predict the positions of the reflectance peaks of the reference sparse FBG sensors from the reflectance spectrum of the dense FBG sensor. We show that a convolution neural network is able to predict the positions of FBG reflectance peaks of sparse sensors with mean absolute error of 7.8 pm that is slightly higher than the hardware reused interrogator equal to 5 pm. We believe that dense FBG sensors assisted with DNNs have a high potential to increase spatial resolution and also extend the length of a fiber optical sensors.

## 1. Introduction

Fiber bragg grating sensors are widely used for distributed temperature and strain sensing [1,2,3]. The performance of FBG-based sensors significantly depends on the accuracy of the peak detection algorithms that provide the possibility of converting a registered signal into temperature/strain values. Commonly, an array of FBGs is sparsely inscribed on a stretch of fiber in spatial and spectral domains to avoid mutual interference between neighboring FBGs. In this case, the peaks positions and their shapes are relatively simple to distinguish and plenty of algorithms may be applied for processing [4,5,6].

However, there is always a trade-off between peak detection accuracy and the number of FBG sensors in the sensing channel. The limited capacity of a sensing channel results in the limited length and spatial resolution of an FBG sensor. Increasing the number of sensing channels limits the sensor interrogation rate and increases the cost of a final device.

At the same time, FBG sensors with high spatial resolution are attractive for various applications including damage processes monitoring [7], healthcare [8], and heat localization [9]. The spatial resolution of a fiber optical sensor may be raised by implementation of chirped FBGs [10] or the implementation more advanced interrogation technique [11].

Software solutions are mainly focused on the coupling of the reflectance spectra of several FBG sensors into one sensing channel and processing the resulting signal for peaks discrimination [12,13]. Today, the scientific community is paying significant attention to machine-learning (ML) algorithms, which have already shown good performance at various fundamental levels and practical applications [14,15]. Particularly, data-driven algorithms are capable of operating with large-scale high-dimensional data and finding hidden intrinsic features and dependencies. There have already been successive attempts to apply ML algorithms for interpretation the of overlapped reflectance spectra from sparse FBG sensors including: extreme learning machines [12], least squares support vector regression [16], convolutional neural networks [17], particle swarm optimization algorithms, long short-term memory algorithms [18] and others [19,20].

However, in most of the presented works, the performances of the algorithms were demonstrated on the model spectra of FBGs, where various additional spectrum distortions associated, for example, with mutual interference between neighboring FBGs, are not considered. Additionally, FBG arrays with a small number of gratings (up to 4 FBGs) were used in experiments, while a real network of sensors can contain more than 50 individual sensing points, for which the presented algorithms will have a significant root mean square error and a significantly longer signal processing time (~s), which makes it difficult to use these algorithms for real-time measurement applications. The possibility of interrogating a sensors network containing 60 FBGs was recently demonstrated [20], but in this case spectral bandwidth was divided into 30 independent regions without crosstalk containing only two paired FBGs with spectral overlap. Each FBG has a bandwidth of ≈0.25 nm to ensure the absence of crosstalk between adjacent regions (1–3 nm), so a length of FBG was ~5 mm, which limits the spatial resolution of measurements. In the case of short FBGs (<1 mm) used for high spatial resolution measurements, the spectral width is much higher (~1.5 nm) and for this reason the spectra will significantly affect each other and therefore these algorithms will give a large error.

Here, we investigate alternative method to process experimental reflection spectra of a highly dense FBG temperature sensor. Fifty closely inscribed FBGs allowed us to increase the length and spatial resolution of the temperature sensor interrogated by using a single optical channel of interrogator. To fabricate the sensor, we used the femtosecond point-by-point inscription technique, allowing high-precision FBG positioning and wavelength resonance specification. For adequate interpretation of a complex reflection signal we do not solve the peak detection problem, but apply deep learning algorithms in order to match the whole reflectance spectrum of the dense FBG sensor with temperature distribution. We also propose an experimental setup based on optical interrogator and Peltier cells for training procedures of deep learning algorithms. By applying the developed algorithms, we show that the capacity of the optical channel of the interrogator can be increased without significant loss of accuracy in the FBG peak detection.

## 2. Experimental Setup

In the following experiments we used two different types of FBG-based temperature sensors inscribed in Fibercore SM1500(9/125)P polyimide-coated fiber by using femtosecond IR laser pulses [21]. The first one was a highly dense sensor composed of 50 uniform FBGs equidistantly arranged along 50 mm fiber segment. Each of the gratings had a length of 0.95 mm and was separated from the neighboring by 0.05 mm, as shown in Figure 1. The resonant wavelengths of the FBGs in the array were chosen to uniformly fill the spectral range of the used 8-channels HBM FS22-SI interrogator (1500–1600 nm). The interrogator operated in optical spectrum analyzer mode providing 1 Spectrum per second refresh rate, 20,001 points per spectrum, and 5 pm resolution. The second type of FBG array was an array containing only intermediate elements of a highly dense array, as shown in Figure 1. Due to the decrease in the FBG density, the reflection spectra the array possess a less noisy shape, making it possible to use such a sensor as a reference when processing the data of a highly dense sensor.

Figure 2a shows a fragment of a reflection spectrum of the highly dense FBG temperature sensor and one of the sparse FBG temperature sensors measured by the optical interrogator. The whole reflectance spectrums are depicted in Figure 2b. As can be seen from the spectra, the close spectral arrangement of the FBGs in a highly dense array leads to mutual interference between adjacent gratings, which consequently increases the noise level of the resonance peaks compared to a sparse FBG array.

We used five calibrated sparse temperature sensors and additional channels of the interrogator to calibrate the dense FBG sensor. The general scheme of the experimental setup is depicted in Figure 3.

FBG sensors were glued closely to each other (as shown at Figure 1) on an aluminum plate with attached Peltier cells on the back surface. By controlling the current and polarity of the electrical power supplies, as well as the positions of Peltier cells, we applied different temperature gradients to the plate, some of which are presented on Figure 4. The temperature of the plate varied in the range from 10 to 80 °C with spatial gradients in the range from −0.38 °C/mm to 0.44 °C/mm. The diversity of the temperature gradients was exploited during the training procedure, thus improving the ability of the deep neural network to generalize incoming data and improve the sensitivity of the dense FBG sensor.

## 3. Architectures of Deep Neural Network

The task for deep neural networks was to predict the reflectance peak positions of 50 FBGs contained in 5 sparse sensors by the optical reflectance spectrum of the highly dense sensor. We investigated the performances of full-connected and convolutional neural networks, which were built using the TensorFlow software package [22]. A fully connected neural network (FCNN) was selected as the most common and simple architecture. Our FCNN consisted of input, hidden and output layers, as shown in Figure 5. The size of the input layer was 20,001 neurons corresponding to the size of the signal array from the interrogator, the output layer had 50 neurons corresponding to the number of the FBGs of the dense sensor. The size of the hidden layer was optimized in order to reduce computational time and maintain the precision of the algorithm. During the grid search procedure of hyperparameters of FCNN we used Mean Squared Error as a loss function and Adamax optimizer for neuron weights optimization. FCNN with 500 neurons at the hidden layer with sigmoid activation function showed the best performance (Section 4).

We also chose a convolutional neural network (CNN) for the task, because of its potential capability to reveal hidden features related to interference between reflectance patterns of nearby FBGs (Figure 6). The input layer was a two-dimensional array of the reflectance pattern, after which there was a layer with one convolutional filter of size 2 × 2 and Relu activation function. The 2-dimensional convoluted image was then flattened and transferred to two fully connected layers. The size of the last output layer was 50 neurons.

Unlike FCNN, where we have used a 1-dimensional array as an input, we transformed the input signal array into a two-dimensional image. This was undertaken in order to reduce the size of a convolutional filter. Our first attempt was to slice 1-dimensional into 50 windows with centered reflectance peaks of FBGs (Figure 7a). However, this approach leads to greater challenges during signal processing due to the aperiodic arrangement of the peaks. Instead, we sliced the input signal into 59 parts, since 59 is a prime factor for 20,001 (Figure 7b). Despite the image losing its physical meaning, such an approach is more robust and straightforward.

The input data was scaled for both neural networks in following order: first, each sample of a reflectance spectra was shifted by mean value of the intensity, then it was normalized by standard deviation of the intensity. Sample size of training, validation and testing datasets were 2000, 500 and 1000, respectively.

## 4. Results

The learning curve of the FCNN on a training dataset and prediction error on a validation dataset are depicted at Figure 8a. We plotted the curves on a logarithmic scale for the convenience of their analysis. During training epochs loss function (mean absolute error) drops down and saturates after 100 epochs. The evolution of the prediction error (orange line) shows that the neural network was not overfitted. Figure 8b shows mean absolute error rates for different activation functions and number of nodes at the hidden layer. Figure 8c demonstrates the predicted positions of the reflectance peaks for all 50 FBGs of the sparse sensors against measured ones in scaled units. Figure 8c is the same curve for a single FBG in nm units. The curves are in close proximity to the straight line corresponding to the ideal case when predicted values are equal to measured values. We found out that FCNN is able to predict the positions of the reflectance peak of the sparse FBG sensors with a mean absolute error equal to 10.9 pm. The root mean square error (RMSE) was 18 pm and the coefficient of determination (R2) was 0.9988.

In the same way we analyzed the performance of CNN (Figure 9). It can be seen that neural networks have similar performance; however, CNN shows the lowest mean absolute error, equal to 7.4 pm, RMSE equal to 14 pm and R2 equal to 0.9993.

The RMSE metric is more sensitive to large errors comparing to MAE. It is clearly seen from Figure 8c and Figure 9c that the mismatch between real, predicted and measured values is not uniformly distributed along different FBGs. For some FBGs, the mean absolute error does not exceed 5 pm; however, for one FBG the mean absolute error reaches 14 pm. We attribute this to the violation of the uniformity of the temperature field across fiber sensors during the mechanical translation of the Peltier cells. Indeed, CNN performed worse at convex temperature distribution when only one Peltier cell was used. The RMSE was equal to 14.48 pm and R2 was equal to 0.9967 for convex temperature distribution comparing to 8.48 pm and 0.9993 for raised gradient temperature distribution. The issue may be solved by adding more Peltier cells with lower size or building more complicated heating/cooling systems, for instance, using laser heating in combination with a spatial light modulator.

Computational complexities of FCNN and CNN may be estimated as follows:(1)$$C_{FCNN}=N_{input}\cdot N_{hidden}+N_{hiden}*N_{output},$$
(2)$$C_{CNN}=N_{input}\cdot D^{2}+N_{input}*N_{fcl1}+N_{fcl1}\cdot N_{fcnn2}+N_{fcnn2}*N_{output},$$
where *N**_inpu_*_t_, *N**_output_*, *N**_hidden_*—numbers of neurons of the input, output layers and hidden layers in FCNN, *D*—dimension of the convolution filter, *N**_fcl1_* and *N**_fcl2_*—numbers of neurons of the fully connected layers in CNN architecture. Calculation of FCNN output takes around 10 million operations, while calculation of CNN output takes around 16 million operations. Better performance of CNN may be related to increased complexity of the architecture. At any case the computational time of the neural networks output is negligible comparing to hardware acquisition time of reflectance spectrum. Computation time of the CNN output from a single sample of the registered reflectance spectrum takes in average 37 milliseconds running on modest graphical processor unit NVIDEA GeForce GTX 950M.

## 5. Conclusions

Thus, the calibration method of a highly dense FBG temperature sensor is proposed in the paper. It provides a possibility for increasing the spatial resolution of a fiberoptic sensor, avoiding the complications of FBG manufacturing or of an interrogation setup. The method is an alternative to the more common approach, wherein several sparse FBGs sensors are coupled into one optical channel. It was shown that deep learning algorithms are capable of mapping the complex reflectance spectrum of the dense sensor with 50 peaks to position of reflectance peaks of the sparse calibrated FBG temperature sensors. The relatively simple architecture of convolutional neural network allowed us to increase the spatial resolution of the dense FBG sensor by five times while maintaining a high temperature resolution close to hardware resolution. Future improvements of the method may be associated with complication of the architecture of the neural network and increasing the uniformity of the temperature distribution across fiber sensors.

## Figures and Tables

**Figure 1 sensors-21-06188-f001:**
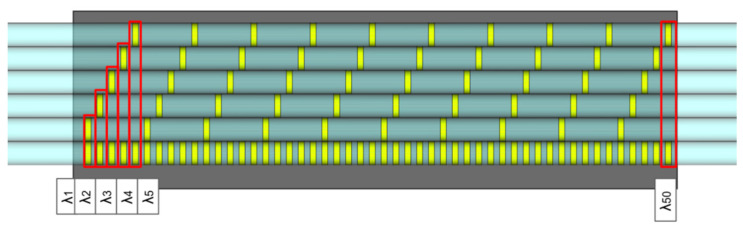
Arrangement of FBGs in highly dense and sparse temperature sensors.

**Figure 2 sensors-21-06188-f002:**
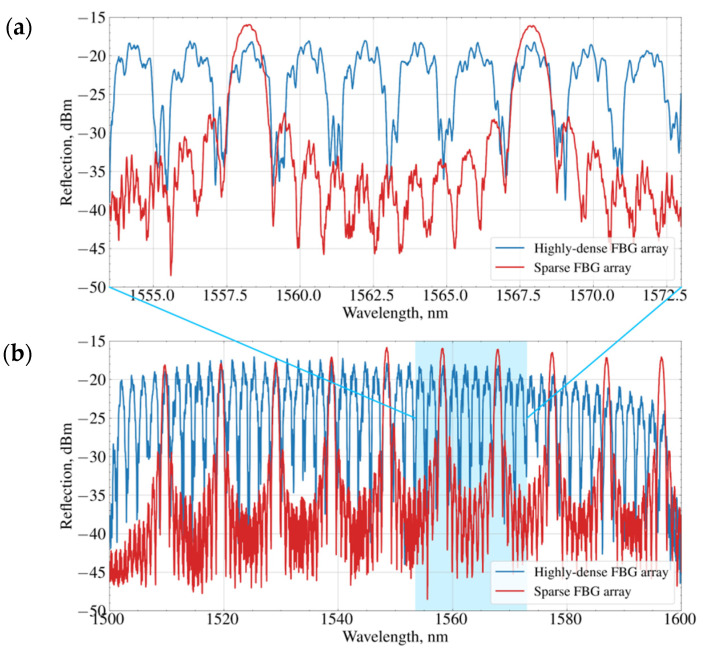
Reflection spectra of a highly dense and one of the sparse FBG sensors: zoomed (**a**) and full **(b)** spectral ranges.

**Figure 3 sensors-21-06188-f003:**
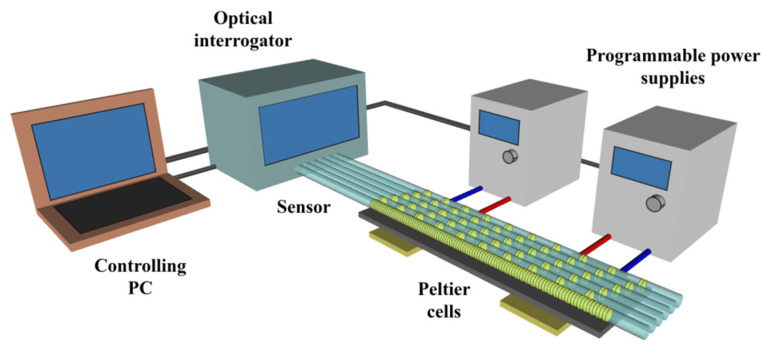
Principle scheme of the experimental setup.

**Figure 4 sensors-21-06188-f004:**
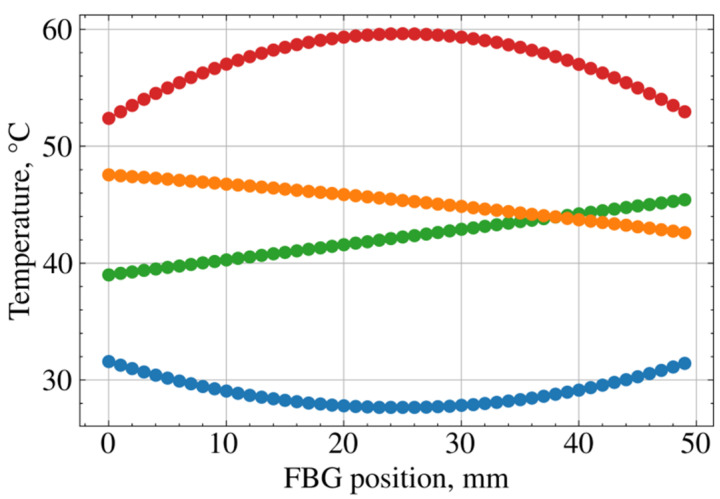
Examples of exploited temperature gradients during training procedure.

**Figure 5 sensors-21-06188-f005:**
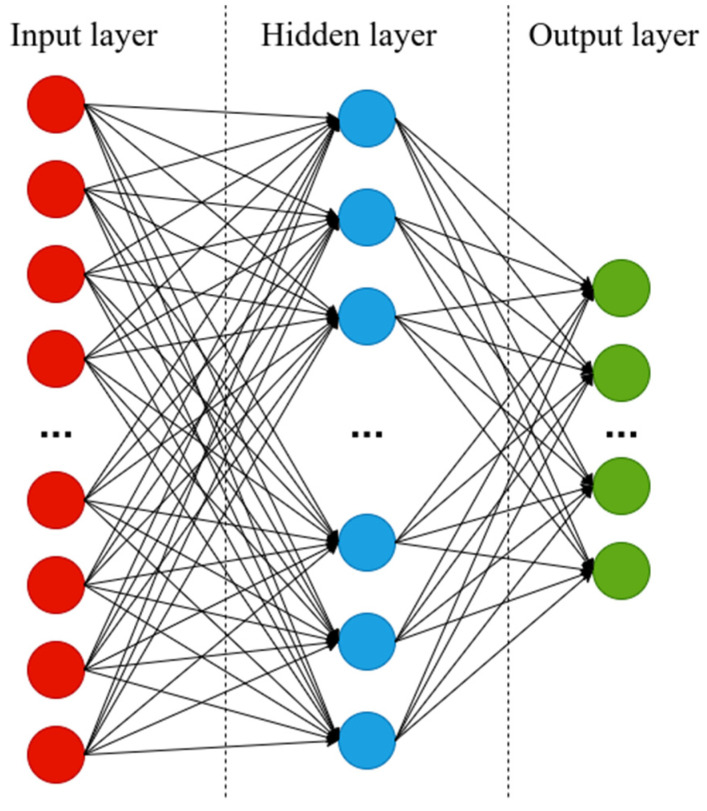
Feed forward neural network with three layers. The final architecture had 20,001 neurons at the input layer, 500 neurons at the hidden layer, 50 neurons at the output layer.

**Figure 6 sensors-21-06188-f006:**
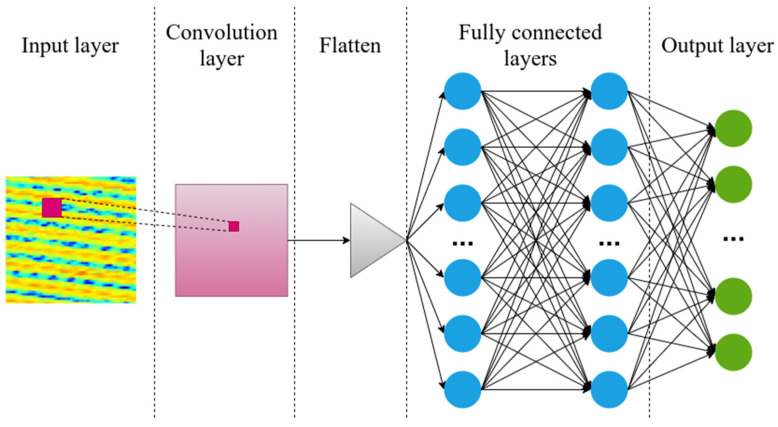
Convolutional neural network consisted of six layers.

**Figure 7 sensors-21-06188-f007:**
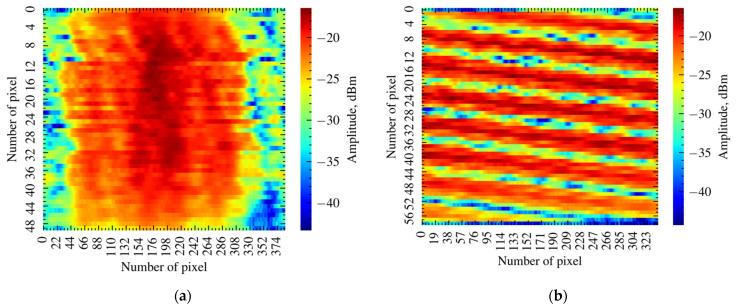
(**a**) 2D reflectance spectrum of FBG sensor with centered reflectance peaks; (**b**) 2D reflectance pattern of FBG obtained by slicing the reflectance spectrum into 59 pieces.

**Figure 8 sensors-21-06188-f008:**
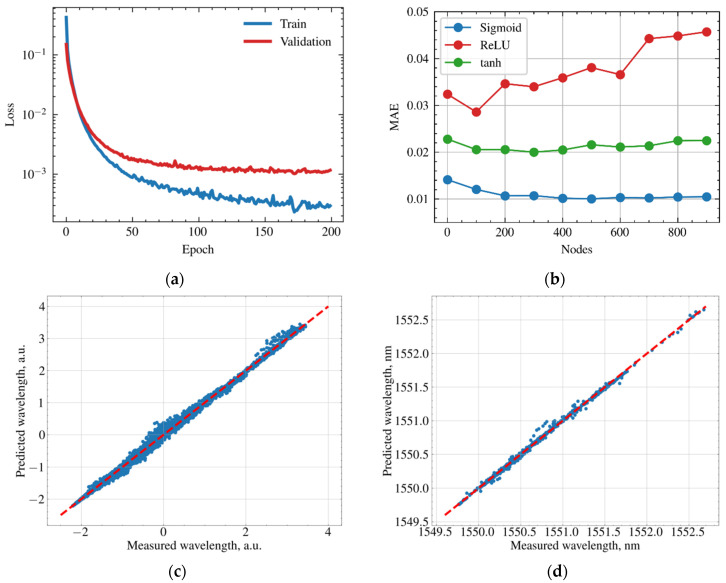
Performance of the FCNN. (**a**) Learning curve of the model on training dataset and evolution of the loss function of validation dataset. (**b**) Mean absolute error of FCNN for different activation functions and different number of nodes of the hidden layer. (**c**) Predicted reflectance peak positions of the FBGs of the sparse sensors against measured values at normalized scales. (**d**) Predicted positions of the reflectance peak for single FBG of sparse sensor against measured value.

**Figure 9 sensors-21-06188-f009:**
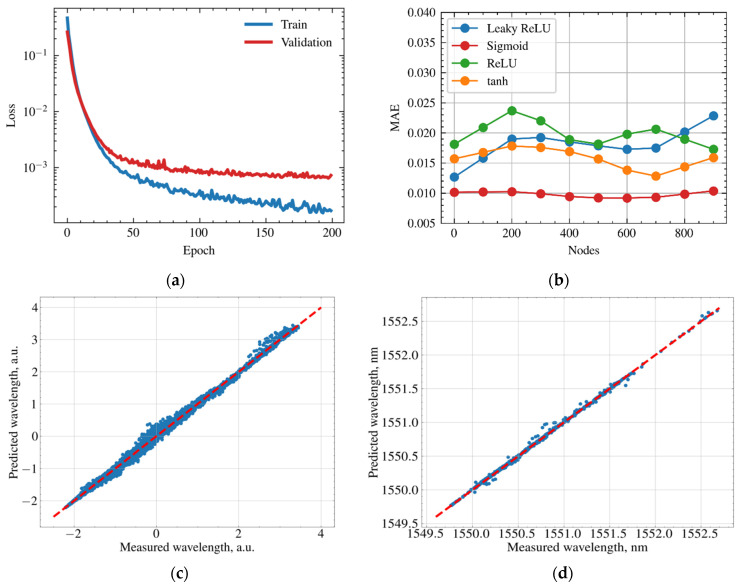
Performance of the CNN (**a**) Learning curve of the model on training dataset and evolution of the loss function of validation dataset. (**b**) Mean absolute error of CNN for different activation functions and different number of nodes of the fully connected layers. (**c**) Predicted reflectance peak positions of the FBGs of the sparse sensors against measured values at normalized scales. (**d**) Predicted positions of the reflectance peak for single FBG of sparse sensor against measured value.

## Data Availability

The data presented in this study is available on request to the corresponding author.

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
