# Peer review of "Highly Dense FBG Temperature Sensor Assisted with Deep Learning Algorithms"

_sensors, 2021, doi:10.3390/s21186188_

Round 1
Reviewer 1 Report
1) The author should elaborate the contribution of the manuscript. What are the existing methods for detecting the peaks of dense FBG sensors, and what is the challenges or disadvantages of those existing methods? The spectrum of dense FBG sensors may show more serious distortion when being used for measure strain, will it possible for strains detection ?
2) The deep learning methods utilized in this manuscript are too simple. Both FCN and CNN are very common for feature extraction. Have you considered to modified the deep neural network as per the tasks of the peak detection. Moreover, the input and output of the two deep neural networks should be elaborated.
3) The deep neural networks are very powerful tools for automatic feature detection. But in the manuscript, only very simple cases have been tested, more complex cases when the temperature distribution of are inhomogeneous should be studied.
4) The manuscript only compared the two deep learning methods used in the manuscript. Comparative studies and analysis with the existing peak detection methods should be provided.
5) The validation experiments are too simple. The study for performance validation of heating and cooling process is highly recommended. And when suffering from inhomogeneous temperatures and stains, the spectrum of the FBG sensors may be distorted or overlapping. what is the performance of the deep learning method under these cases. Will this affect the generalization capacity of the two deep learning methods?
6) There should be typos on page 6 "2 results".
Author Response
We are pleased to see positive comments and we would like to thank referees for the remarks and questions that helped us to improve the clarity of the presentation of the results. We have made corresponding changes to the revised manuscript, in which we have addressed all the questions and suggestions. Below are line-by-line responses to all the comments.
1) The author should elaborate the contribution of the manuscript.What are the existing methods for detecting the peaks of the dense FBG sensors, and what is the challenges or disadvantages of those existing methods? The spectrum of dense FBG sensors may show more serious distortion when being used for measure strain, will it possible for strains detection ?
Contribution of the manuscript is a demonstration of a new method to calibrate dense FBG-based temperature sensors with assistance of DNNs algorithms. We introduce the term “dense FBG sensor” to describe the array of densely inscribed FBGs with distorted reflectance peaks. Distortions of the peaks are provoked by interference phenomena appearing between closely located FBG. We would like to emphasize that dense FBG sensors are commonly avoided in practice due to the difficulty in interpreting the position of the reflectance peaks. We hope that our findings will pave the way towards utilization of such sensors in real applications.
We are not aware of any studies in which the analysis of FBG array spectra with a similar space and spectral density is performed. However, with distributed temperature measurements, a number of alternative approaches are used, which are based on spectral and/or time domain analysis of long chirped FBGs [Tosi, Daniele. 2018. "Review of Chirped Fiber Bragg Grating (CFBG) Fiber-Optic Sensors and Their Applications" Sensors 18, no. 7: 2147. https://doi.org/10.3390/s18072147], or a long uniform FBGs [Juan Sancho, Sanghoon Chin, David Barrera, Salvador Sales, and Luc Thévenaz, "Time-frequency analysis of long fiber Bragg gratings with low reflectivity, "Opt. Express 21, 7171-7179 (2013)]. A key advantage of our approach is the increased resolution and accuracy of the FBG-based sensor when using a standard interrogation unit. Please, see line 40
We also pointed out the challenges or disadvantages of existing methods for detecting the peaks of dense FBG sensors in the introduction of the revised version of manuscript. Please, see line 56.
In principle implementation of DNNs for strain detection is possible, however, the calibration system should be considerably modified.
2) The deep learning methods utilized in this manuscript are too simple. Both FCN and CNN are very common for feature extraction. Have you considered to modified the deep neural network as per the tasks of the peak detection. Moreover, the input and output of the two deep neural networks should be elaborated.
We agree that applied architectures of deep neural networks in our case are relatively simple. Simplicity of DNNs brings two advantages: 1) calculation speed 2) the possibility to use low-performance CPU/GPU.
We expect the need to upgrade the architectures of DNNs with complication of the temperature distributions. However, in our case, when hardware resolution is almost reached, further complication of DNNs architecture is redundant.
The input of DNNs is determined by raw data from the interrogator. Signal processing of the input data, e.g. averaging or convolution, may be implemented after the input layer of DNN. The output of DNNs corresponds to spatial resolution of the dense sensor that is determined by the number of calibration sensors (in our case the number of sparsely inscribed FBGs).
3) The deep neural networks are very powerful tools for automatic feature detection. But in the manuscript, only very simple cases have been tested, more complex cases when the temperature distribution of are inhomogeneous should be studied.
We agree that the major goal for the method is to deal with as much complicated temperature distribution as possible. However, we would like to emphasize that the main focus of this manuscript is to operate with FBG temperature sensors with distorted reflectance spectra caused by densely inscribed FBG. For this reason we chose the “Communication” format of the manuscript. Our future investigations will be devoted to extreme situations including implementation of different strains and significant overlapping of reflectance peaks of different FBGs.
4) The manuscript only compared the two deep learning methods used in the manuscript. Comparative studies and analysis with the existing peak detection methods should be provided.
We would like to emphasize that we do not solve peak detection problem directly with our method. We predict the peak positions of the sparse temperature sensors from entire distorted spectrum of the dense sensor. Sparse sensors may be replaced by thermistor sensors or thermal image camera that do not require peak positions detecting.
One may compare the linearity of the spectral shift of the distorted peak against the temperature. We apply gaussian fit to get the positions of distorted peaks and plot them against spectral shift of the peaks of the sparse sensors. It is clearly seen that gaussian fit shows poor performance (Please see figures at attached pdf file ).
However, adequate comparison of two methods is questionable and we prefer avoid it in the manuscript.
5) The validation experiments are too simple. The study for performance validation of heating and cooling process is highly recommended. And when suffering from inhomogeneous temperatures and stains, the spectrum of the FBG sensors may be distorted or overlapping. what is the performance of the deep learning method under these cases. Will this affect the generalization capacity of the two deep learning methods?
We believe that this question and also 2) and 3) are closely related and we hope that we answered exhaustively.
We also should point out that DNN is effective only if it operates on the validation distribution matched with training distribution. Other words, if we try to predict the temperatures far away from the temperatures that we used to train DNNs, their performances will be poor. The same results we expect in case of strains. To solve the issue, one should manage the calibration system close to requirements of a specific application.
6) There should be typos on page 6 "2 results".
We apologize for the typos made. We now have corrected the misprints in the revised manuscript.

Reviewer 2 Report
In the paper Highly-dense FBG temperature sensor assisted with deep learning algorithms is presented the performance of deep neural networks for processing the reflectance spectrum from a fiber-optic temperature sensor.
I have some suggestion:
The sencences "In this paper, we demonstrate the performance of deep neural networks for processing the reflectance spectrum from a fiber-optic temperature sensor composed of 50 fiber Bragg gratings 10 (FBGs) arranged in the 1500-1600 nm spectral range." can be reformulated.
The abstract can be improved-"The aim for deep neural networks is" in this study the DNN ...
Keywords must be relevant
Line 57 "In the following experiments" can be reformulated
Fig 5 and Fig 6 are not relevant
The conclusion must be improved
Author Response
We are pleased to see positive comments and we would like to thank referees for the remarks and questions that helped us to improve the clarity of the presentation of the results. We have made corresponding changes to the revised manuscript, inwhich we have addressed all the questions and suggestions. Below are line-by-line responses to all the comments.
In the paper Highly-dense FBG temperature sensor assisted with deep learning algorithms is presented the performance of deep neural networks for processing the reflectance spectrum from a fiber-optic temperature sensor.
I have some suggestion:
1) The sencences "In this paper, we demonstrate the performance of deep neural networks for processing the reflectance spectrum from a fiber-optic temperature sensor composed of 50 fiber Bragg gratings 10 (FBGs) arranged in the 1500-1600 nm spectral range." can be reformulated.
We reformulated the abstract. We hope that the new variant stresses the novelty of our work and avoids misunderstanding.
2) The abstract can be improved-"The aim for deep neural networks is" in this study the DNN …
This comment relates to the abstract and we hope that the new variant is acceptable for the reviewer.
Keywords must be relevant
We have added more keywords to be more specific about our work:
fiber Bragg grating, optical fiber sensor, distributed temperature sensor, deep learning algorithms, fully-connected neural network, convolutional neural network
3) Line 57 "In the following experiments" can be reformulated
This expression is quite common, therefore, we would like not change the sentence
4) Fig 5 and Fig 6 are not relevant
In our opinion, these figures demonstrate the architectures of the neural networks used in the work and greatly simplify the understanding of the data processing procedure.
5) The conclusion must be improved
We rewrite the conclusion part by underlying the novelty and the main result of our work.
Reviewer 3 Report
It is a good idea to combine the Highly-dense FBG temperature sensor with the deep learning algorithms and the authors prove that a convolution neural network is able to predict the positions of FBG reflectance peaks of sparse sensors. The manuscript is well organized with easy to understand English. I have some moderate revision suggestion to this article:
(1) The gap for which prediction of position of FBG reflectance peaks should be further stressed, which emphasized the use of ML in this study.
(2) Lines 40-43, some recent relevant papers may be referred to, such as:
for extreme learning
Prediction of undrained shear strength using extreme gradient boosting and random forest based on Bayesian optimization
Efficient reliability analysis of earth dam slope stability using extreme gradient boosting method
for convolutional neural networks
Zhang, W., Li, H., Li, Y. et al. (2021). Application of deep learning algorithms in geotechnical engineering: a short critical review. Artificial Intelligence Review. https://doi.org/10.1007/s10462-021-09967-1
(3) for FCNN, the structure with 500 neurons at the hidden layer with sigmoid activation function showed the best performance, it would be better if the authors can provide the performance comparison of different numbers of neurons and activation functions.
(4) Same suggestion for CNN model.
(5) Can the authors provide the information of performance measures such as the coefficient of determination R2, RMSE to quantify the fitting accuracy from the FCNN and CNN model.
(6) The innovation of this study should be further stressed in the conclusion part.
Author Response
We are pleased to see positive comments and we would like to thank referees for the remarks and questions that helped us to improve the clarity of the presentation of the results. We have made corresponding changes to the revised manuscript, in which we have addressed all the questions and suggestions. Below are line-by-line responses to all the comments:
It is a good idea to combine the Highly-dense FBG temperature sensor with the deep learning algorithms and the authors prove that a convolution neural network is able to predict the positions of FBG reflectance peaks of sparse sensors. The manuscript is well organized with easy to understand English. I have some moderate revision suggestion to this article:
(1) The gap for which prediction of position of FBG reflectance peaks should be further stressed, which emphasized the use of ML in this study.
Please, see the revised introduction section. We emphasized the novelty of our work and advantages that it may bring to the area of fiber optical sensors.
(2) Lines 40-43, some recent relevant papers may be referred to, such as:
for extreme learning
Prediction of undrained shear strength using extreme gradient boosting and random forest based on Bayesian optimization
Efficient reliability analysis of earth dam slope stability using extreme gradient boosting method
for convolutional neural networks
Zhang, W., Li, H., Li, Y. et al. (2021). Application of deep learning algorithms in geotechnical engineering: a short critical review. Artificial Intelligence Review. https://doi.org/10.1007/s10462-021-09967-1
We add the third reference into the introduction section related to machine learning algorithms (Line 56). However, the fist two references are quite far from the scope of our manuscript. Undoubtedly, the works apply deep learning algorithms in geotechnical engineering, but are not related to optics and fiber bragg sensors.
(3) for FCNN, the structure with 500 neurons at the hidden layer with sigmoid activation function showed the best performance, it would be better if the authors can provide the performance comparison of different numbers of neurons and activation functions
We add a figure 8b to show the MAE value of FCNN for different numbers of neurons and activation function.
(4) Same suggestion for CNN model.
We add the graph of MAE evolution of CNN for different activation function optimization and nodes of fully connected layers by analogy with FCNN (Fig 9b). However, optimization of CNN is a high-dimensional problem. Therefore, one needs to perform a massive amount of graphs that will overload our manuscript. We prefer to list optimal parameters of CNN that were found.
(5) Can the authors provide the information of performance measures such as the coefficient of determination R2, RMSE to quantify the fitting accuracy from the FCNN and CNN model.
RMSE metric is more sensitive to large errors compared to MAE. For FCNN RMSE was 18 pm and R2 was 0.9988. For CNN RMSE was 14 pm and R2 was 0.9993.
We attribute this to the violation of the uniformity of the temperature field across fiber sensors during mechanical translation of the Peltier cells. Indeed, CNN performs worse at convex temperature distribution when only one Peltier cell was used. The RMSE was equal to 14.48 pm and R2 was equal to 0.9967 for convex temperature distribution comparing to 8.48 pm and 0.9993 for raised gradient temperature distribution.
We add the discussion to the manuscript, please see line 173 and 182.
(6) The innovation of this study should be further stressed in the conclusion part.
We have reformulated the conclusion part by stressing the novelty and difference between the existing methods.
Round 2
Reviewer 1 Report
The authors tried to replied all my comment, and the manuscript has been improved a lot. It can be accepted for publication now.
Reviewer 2 Report
The paper Highly-dense FBG temperature sensor assisted with deep 3
learning algorithms, can be published in the presented form.